# MapperPlus: Agnostic clustering of high-dimension data for precision medicine

**Esha Datta[1], Aditya Ballal[2], Javier E. López[3], Leighton T. Izu[2]\***

**1** Department of Mathematics, Graduate Group in Applied Mathematics, University of California, Davis, United States of America, **2** Department of Pharmacology, University of California, Davis, United States of America, **3** Department of Internal Medicine, Division of Cardiovascular Medicine, and Cardiovascular Research Institute, University of California, Davis, United States of America

\* ltizu@ucdavis.edu

## Abstract

One of the goals of precision medicine is to classify patients into subgroups that differ in their susceptibility and response to a disease, thereby enabling tailored treatments for each subgroup. Therefore, there is a great need to identify distinctive clusters of patients from patient data. There are three key challenges to three key challenges of patient stratification: 1) the unknown number of clusters, 2) the need for assessing cluster validity, and 3) the clinical interpretability. We developed MapperPlus, a novel unsupervised clustering pipeline, that directly addresses these challenges. It extends the topological Mapper technique and blends it with two random-walk algorithms to automatically detect disjoint subgroups in patient data. We demonstrate that MapperPlus outperforms traditional agnostic clustering methods in key accuracy/performance metrics by testing its performance on publicly available medical and non-medical data set. We also demonstrate the predictive power of MapperPlus in a medical dataset of pediatric stem cell transplant patients where a number of cluster is unknown. Here, MapperPlus stratifies the patient population into clusters with distinctive survival rates. The MapperPlus software is open-source and publicly available.

## Author summary

The era of precision medicine represents a unique and exciting opportunity in transforming the way we treat patients. With the immense availability of biomedical data and new computational techniques, we are more able than ever to understand what makes a patient unique. Indeed, even for a single condition, we can recognize that there are heterogeneities within the patient population. Understanding these differences can and should influence the way we treat patients. Key to this process is patient stratification, which is the division of patient populations into clinically meaningful subgroups. The goal of patient stratification is to capture the individuality of patients without becoming overly fine-grained. This is an exciting balancing act that engages both meaningful medical and mathematical questions. We develop the MapperPlus pipeline for patient stratification. This is an unsupervised learning pipeline that leverages the mathematical notion of topology to

**Data Availability Statement:** All datasets used in this work are publicly available in the UCI Machine Learning Repository (https://archive.ics.uci.edu/). The links for the datasets are given below: Wine Dataset: https://archive.ics.uci.edu/dataset/109/

wine. Breast Cancer Dataset: https://archive.ics.uci.edu/dataset/17/breast+cancer+wisconsin+diagnostic Iris Dataset: https://archive.ics.uci.edu/dataset/53/iris Rice Dataset: https://archive.ics.uci.edu/dataset/545/rice+cammeo+and+osmancik Bone marrow Dataset: https://archive.ics.uci.edu/dataset/565/bone+marrow+transplant+children The open-source software described in this work can be found at https://github.com/lordareicgnon/Mapper_plus.

**Funding:** ED was supported by T32 Training Program for Institutions That Promote Diversity HL 086350. LTI and AB were supported by National Institutes of Health (NIH)/National Heart, Lung, and Blood Institute (NHLBI) R01HL149431. LTI and JEL were supported by NIH/NHLBI U01 HL 160274. The funders had no role in study design, data collection and analysis, decision to publish, or preparation of the manuscript.

**Competing interests:** The authors have declared that no competing interests exists.

detect clusters within high-dimensional data. It is effective in many settings and we demonstrate, in particular, its efficacy in a precision medicine application.

# 1 Introduction

Precision medicine research aims to improve clinical outcomes of individual patients by developing treatments on the basis of characteristics which make them unique [1]. Integral to this approach is analyzing and understanding patient data; integrating biomarkers, genetic, phenotypic, and epigenomic data allows for a deeper understanding of the patient. By understanding what makes each patient unique, we can tailor the optimal treatment.

Yet this desire to individualize patients represents a paradigm shift away from the historical practice of medicine. Medical treatments are typically designed for the so-called "average" patient [2]. This design is born out of necessity: evidence-based medicine requires that patients are grouped for clinical studies and the statistical testing of medical hypotheses [1]. Thus we arrive at one of the central challenges of precision medicine: how does one treat each patient as an individual when robust scientific practices are developed by the grouping of patients?

Patient stratification is "the division of a patient population into distinct subgroups based on the presence or absence of particular disease characteristics" [3]. The goal is to develop clinically meaningful subgroups of patients within a single diagnosis or conditions of interest. This is especially relevant for highly heterogeneous diseases with multiple comorbities, such as cardiovascular diseases [4–6]. There are numerous computational methodologies available for patient stratification, including clustering, dimensionality reduction, similarity measures, and supervised learning techniques [1]. In a 2018 systematic review of patient similarity analysis for precision medicine, it was found that clustering was the most popular approach for analyzing high-dimensional data for subtyping disease [1]. Clustering was used when the the researchers' primary goal was to define groups of patients who had a "similar disease evolution" [1].

Clustering refers to the process of identifying subgroups within multidimensional data on the basis of some notion of similarity [7]. The process of clustering typically requires mathematically defining a measure of similarity in order to create subgroups. Our long-term goal is to better define clustering in the context of precision medicine. To that effect, we identify three key challenges that must be addressed in order to apply clustering to patient stratification.

First, it is not known how many subtypes exist within a single medical condition. For instance, heart failure, a disease for which two main subtypes have been characterized: preserved ejection fraction and reduced ejection fraction. And yet, recent research has characterized a "grey-zone" between these two types [8] which may represent a different subtype of the disease. Within precision medicine, there is an ongoing need to better detect subtypes from complex pathophysiologic states [9]. Thus, clustering approaches for patient stratification which rely on the *a priori* knowledge of the number of clusters can be inherently flawed. They rely on an initial assumption of the number of subtypes, even when such information may not be available. We emphasize this because popular clustering techniques such as k-means and k-medoids require the user to input the number of clusters [10]. These techniques do have some heuristics by which the optimal number of clusters can be estimated, such as the elbow method, but such methods do not always provide a clear answer in many situations [11]. These clustering methods, while efficacious in the abstract, necessitate some degree of insight that may not be available. We propose that agnostic approaches that do not require such user

input are better suited to biomedical applications to heterogeneous and multi-factorial diseases.

Second, we must have a method for assessing cluster validity. Many clustering methods require the selection of parameters by the end user [9] and every parameter set can return a clustering. The onus is on the researcher to determine which of these clusterings, if any, are acceptable and there is ongoing research in measuring the validity of clustering results [12]. We refer to the decision of choosing the appropriate parameter set for a given clustering application as the problem of parameter selection. We argue that parameter selection is especially relevant for patient stratification, as the subgroups detected may be used to guide future medical treatment. Thus, clustering techniques for precision medicine applications necessitate a rigorous and reproducible heuristic for cluster validation.

Finally, the clusters that are detected must be clinically interpretable. Interpretability is a major consideration of computational models; a model must yield results that are understandable to its users [9]. For the patient stratification problem, we must be able to divide patients not only into mathematically valid partitions, but also into clinically meaningful subgroups [1]. Moreover, it would be advantageous to be able to statistically test the characteristics of these groups. Such methods are necessitated by standard evidence-based medical practices [1]. As such, clustering techniques that return large numbers of clusters each with very few patients may not be interpretable in the medical context. A precision medicine clustering algorithm must be sensitive enough to detect meaningful differences between patients, but not so sensitive as to become difficult to interpret.

Topological data analysis (TDA) refers to a broad set of tools for data exploration and analysis that leverage topology, a field of mathematics that rigorously examines notions of shape and connectivity [9]. TDA is an emergent field with growing popularity in the biomedical sciences that we propose can address some of these challenges. It has a unique approach to data and can often uncover patterns that traditional approaches cannot. It can detect multivariate shape-based features, both local and global behavior, and address the so-called "curse of dimensionality" [9]. While a wealth of TDA techniques exist, we focused on Mapper, a particular TDA algorithm that may have several advantages over traditional statistical and machine learning techniques.

Mapper [13] simplifies and visualizes high dimensional data as a graph using topology. To do this, Mapper takes in high-dimensional data, projects it to a lower dimension, bins the projected data in an overlapping manner, and then uses that projection to generate a graph but not clusters. Often, flares (long extended branches) and loops (connected circular figures) in the Mapper graph correspond to interesting subpopulations in a dataset [14]. Mapper has proven to be a popular tool for precision medicine research [9] with applications to conditions like aortic valvular stenosis [15], attention-deficit/hyperactivity disorder [16], and traumatic brain injury [17]. In many of these cases, Mapper has been used to identify subtypes of patients within each individual condition.

The major roadblock for this technique is that Mapper alone cannot cluster data. Clustering requires the identification of subgroups [7] and the final output of Mapper is a graph. Dividing the resultant graph into distinct parts is not sufficient for partitioning the data into disjoint groups. Due to the overlapping nature of the bins, individual data points can belong to multiple nodes in the graph, leading to clusters that may share observations [18]. The only scenario in which this approach produces completely disjoint clusters is if the mapper-graph contains multiple connected components, and the optimal clustering of the graph aligns with these connected components.

This problem has been approached by researchers in different ways. Some simply use the overlapping clustering [16], while excluding the datapoints that appear in more than one

cluster from the analysis [19]. None of these approaches address the problem of patient stratification. An overlapping clustering corresponds to some patients being assigned to more than one treatments which is not possible. Similarly, patients must not be excluded since this would introduce bias. Thus there is a need for mew techniques that can convert the Mapper graph output into disjoint clusters.

Here, we present MapperPlus, a novel unsupervised clustering pipeline which addresses this need. MapperPlus builds on Mapper's topological graph and blends it with two random-walk techniques [20] for clustering on a graph network. MapperPlus automatically detects the number of clusters within a dataset and then generates a disjoint clustering of the data. This is a novel extension of the Mapper algorithm, allowing us to convert Mapper's topological information into disjoint subgroups. This extension has been developed specifically with the needs of precision medicine in mind: we show how MapperPlus directly addresses the three key challenges of clustering in non-medical data sets or medical sets for tissue characterization or patient stratification.

In this report, we first present a brief description of the Mapper algorithm and two random-walk techniques: the walk-likelihood algorithm and the walk-likelihood community finder. Then, we introduce the MapperPlus pipeline. This 8-step process generates the Mapper graph and agnostically computes disjoint clusters based on the topology of the data. To test the general efficacy of the pipeline, we compared its clustering against other algorithms on multiple publicly available datasets including a breast cancer tissue dataset. Finally, in a patient stratification case study, we apply MapperPlus to a dataset of pediatric stem cell transplant patients. We find that MapperPlus has significant utility for patient stratification by showing differences in survival rates and clinical characteristics of included subjects.

## 2 Methods

### 2.1 Mapper

Mapper is a TDA algorithm that leverages the geometric properties of a dataset in order to generate a graph. It operates by defining a lens (which can be any function) on the dataset and then clustering at different ranges of values of this lens while preserving the local connectivity of those clusters. It takes a dataset as input and returns a graph $G_M(V_M, E_M)$, where vertices $V$ represent bins of observations and edges $E$ are placed between vertices that share one or more observations. Table 1 lists the frequntly-used symbols and their meaning.

Consider a dataset, which can be represented as a finite metric space $(X, \delta_X)$. The dataset will be projected to a lower dimension via a lens (or filter function) $f: X \to Y$ where $Y \subset \mathbb{R}^m$. The choice of lens is critical and often dependent on the dataset.

**Table 1. Table of frequently-used mathematical notation.**

| | |
|---|---|
| $G_M(V_M, E_M)$ | Mapper graph with vertex set $V_M$ and edge set $E_M$ |
| $X$ | Metric space equipped with metric $\delta_X$ |
| $f$ | Filter function or lens |
| $\mathcal{C}$ | The cover, consisting of overlapping hypercubes |
| $C_n$ | The $n$th hypercube of the cover |
| $r$ | Resolution parameter for the cover, may be any natural number |
| $g$ | Gain parameter for the cover, a percentage such that $0\% < g < 100\%$ |
| $G$ | Network of instances |

Mapper takes two parameters as input, resolution $r$ and gain $g$, which determine the cover. The resolution determines how fine-grained the examination of the dataset will be, while gain tunes the connectivity of the points in the dataset. The resolution and gain are used by the Mapper algorithm as follows. A cover $\mathcal{C} = \{C_n\}$ is constructed on the lens $f$. The cover is typically taken to be a collection of $r^m$ overlapping hypercubes, where $m$ is the dimension of the lensed space. The resolution determines the number of hypercubes used in the cover, while the gain determines the extent to which the hypercubes overlap. We can think of the resolution as "binning" the range of the lens.

Mapper then pulls back the cover, which is to say, it identifies which of the points fall into each hypercube $C_n$ under the function $f$. That is, we look at the inverse image $f^{-1}(C_n) \subseteq X$ for each $C_n \in \mathcal{C}$. Crucially, some points will fall into more than one inverse image, which is a facet of the gain. Intuitively, the higher the gain, the more points will be shared by multiple inverse images.

At this point, Mapper applies a clustering method within each inverse image $f^{-1}(C_n)$. This can be thought of as a partial clustering. The clustering method is never applied to the dataset as a whole. Rather, it is applied to subdivisions of the dataset as determined by the hypercubes $C_n$. Each cluster in each inverse image forms a node. Mapper then preserves the local connectivity of the clusters by drawing an edge between two nodes if the corresponding clusters share an observation. In this manner, Mapper produces a sparse representation of the original dataset.

## 2.2 Walk-likelihood algorithm

The walk-likelihood algorithm (WLA) is a graph partitioning algorithm first introduced by Ballal, Kion-Crosby, and Morozov [20]. The algorithm takes the transition matrix of a graph, the connectivity of each node in the graph, and an initial guess, a partition of the graph into $m$ communities. Beginning from the initial partition, WLA iteratively computes the number of times per random walk that a node $n$ is visited by random walks with $l_{max}$ steps which start from node $n'$ in community $c$. The assignment of nodes to communities is then updated in a Bayesian manner using this probability. The process is repeated until the iterative node reassignment process converges to a stable assignment.

## 2.3 Walk-likelihood community finder

The walk-likelihood community finder (WLCF) is a community detection algorithm introduced in [20] that exploits the random walk properties of graphs to detect communities. Unlike WLA, WLCF automatically detects the number of communities within a graph based on the modularity, a quantitative metric that evaluates the degree of community structure in a network [21], of the graph. It does not require any a priori knowledge of the number of communities for its computation.

WLCF functions by first bifurcating a graph into two communities. Within those two communities, WLA is then applied to obtain a more accurate graph partition. The modularity score is computed for all $m$ communities and pairs of communities are merged together if this would increase the modularity. This process of bifurcation and merging repeats until the number of communities in the partitions obtained by subsequent iterations remains constant and converges to a stable community assignment.

## 2.4 MapperPlus

Our pipeline proceeds in 8 distinct stages. Each major step is explained in detail below, with a flowchart of the pipeline shown in Fig 1. Typically, data inputted to MapperPlus is normalized, for example, converted to $z$-scores.

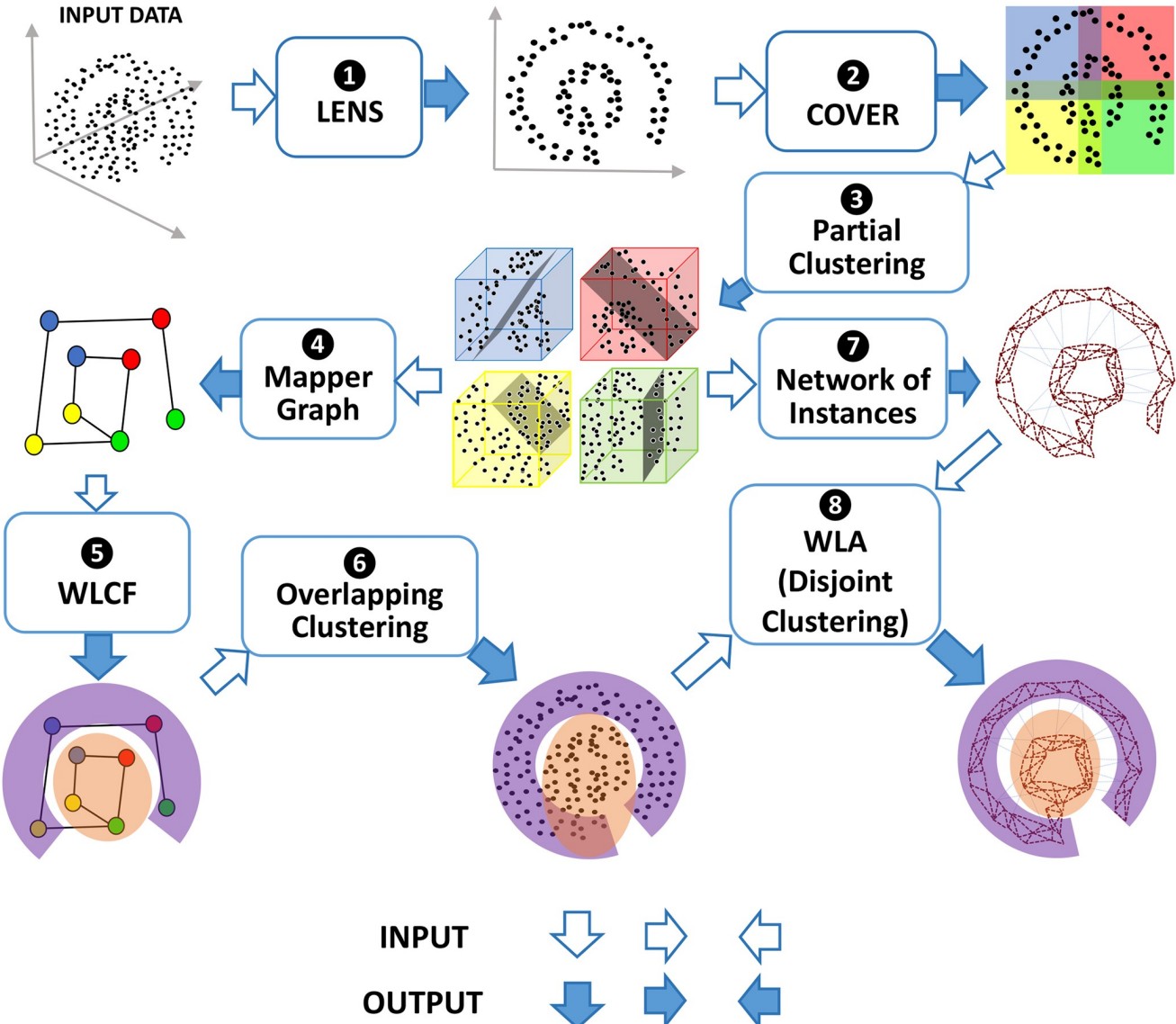

**Fig 1. The MapperPlus pipeline can be divided into 8 distinct steps, beginning with projecting the dataset to a lower-dimension space via a lens and ending with a disjoint clustering of the dataset.**

1. **Define a lens**: Define a function $f: X \to Y$, which maps the dataset to a lower dimension, where $Y \subset \mathbb{R}^m$.

2. **Define a cover**: Select a resolution $r \in \mathbb{N}$ and a gain $g \in [0, 1]$. A cover $\mathcal{C} = \{C_n\}$ is constructed on lensed data using $r^m$ hypercubes with $g$ overlap in each dimension. Each $C_n$ is a hypercube and the union $\bigcup_n C_n$ contains $Y$.

3. **Cluster each inverse image**: Use a clustering algorithm to cluster observations in the inverse image $f^{-1}(C_n)$ of each hypercube.

4. **Generate Mapper graph**: Generate a graph $G_M(V_M, E_M)$ where each node is a collection of not-necessarily-distinct observations and edges $e_M$ connect nodes that share observations.

The edges $e_M$ are weighted by the number of observations the nodes share. Note that the nodes in the graph $G_M(V_M, E_M)$ are connected to themselves.

5. **WLCF on Mapper graph**: We apply WLCF to the Mapper graph $G_M$, which yields a partition of $G_M$ into $m$ communities. This automatically determines the optimal number of communities present within the dataset.

6. **Overlapping clustering**: WLCF yields $c$ **overlapping** communities of instances. This is done by taking the union over all nodes within each detected community. The overlap is due to the fact that the Mapper nodes need not contain distinct observations.

7. **Network of instances**: At this crucial step, we translate the information in the Mapper graph $G_M(V_M, E_M)$ into a new graphical representation $G$ as follows:

   1. For each observation $i$ in the dataset, create a node $n_i$.

   2. For distinct observations $i, j$, draw an edge $e_{ij}$ between their corresponding nodes $n_i$ and $n_j$ if $i, j \in n_M$ for some node $n_M \in V_M$. That is, we connect two nodes in the new graph representation if the corresponding instances would be binned together in the original Mapper graph. The edge $e_{ij}$ is weighted by the number of such nodes $n_M$ that exist. Note that we also include self-edges for the network of instances ($e_{ii}$).

   3. For distinct observations $i, j$, draw an edge $e_{ij}$ if $i, j \in f^{-1}(C_n)$ for some $n$. Now we are connecting instances if they fall in the pullback set of the same hypercube under the Mapper construction. Suppose there are $k$ instances in the hypercube $C_n$. Then the edge $e_{ij}$ is weighted by $\frac{1}{k}$.

      This step is crucial because it translates the information embedded within the overlapping nodes of a Mapper graph into a non-overlapping network of individual instances. The manner in which individual instances are connected preserves all of the topological information inherent to the Mapper graph while resolving the problem of the overlapping nodes.

8. **WLA (Disjoint clustering)**: In order to obtain a final clustering on the novel graph $G$, we apply WLA with $c$ communities (the number of communities having been automatically detected in step 5). This yields a disjoint clustering of the original dataset into $c$ clusters.

To summarize, the original Mapper algorithm described by steps 1–4 of this pipeline converts the data into Mapper graph; a graph that embeds the topological features of the data. This graph is clustered into communities (Step 5) to obtain overlapping clusters (Step 6). We further use the information on Mapper graph and overlapping clusters to obtain the graph of instances (Step 7) and cluster them into disjoint clusters (Step 8). Steps 5–8 represent our extension of the Mapper algorithm.

## 2.5 Software availability

An open-source Python implementation of MapperPlus is available for use.

## 2.6 Normalized mutual information

MapperPlus relies on two tunable parameters, gain and resolution, which affect the resultant clustering. To compare distinct sets of clusters, we use normalized mutual information (NMI). The NMI compares two sets of clusters and returns a score between 0 and 1. Consider a dataset

of $N$ observations and a clustering $U_{N \times m}$ of the observations into $m$ communities, where $U_{ic} = 1$ if and only if node $i$ belongs to community $c$. Given two such clusterings $U_{N \times m}$ and $U'_{N \times m'}$, the normalized mutual information between them is defined as ([22]):

$$\text{NMI}(U, U') = \frac{2\text{I}(U, U')}{\text{I}(U, U) + \text{I}(U', U')}, \tag{1}$$

where I is the mutual information between $U$ and $U'$ defined as

$$\text{I}(U, U') = \sum_{c=1}^{m} \sum_{c'=1}^{m'} P_{UU'}(c, c') \log \frac{P_{UU'}(c, c')}{P_U(c) P_{U'}(c')}, \tag{2}$$

where $P_U(c)$ is the probability that if an observation $i$ is chosen at random from the dataset, it belongs to cluster $c$ of the clustering $U$, given by

$$P_U(c) = N^{-1} \sum_{i=1}^{N} U_{ic}, \tag{3}$$

and $P_{UU'}(c, c')$ is the probability that if an observation $i$ is chosen at random from the dataset, it belongs to both the cluster $c$ of the clustering $U$ and the cluster $c'$ of the clustering $U'$, given by

$$P_{UU'}(c, c') = N^{-1} \sum_{i=1}^{N} U_{ic} U'_{ic'}. \tag{4}$$

Note that because of the normalization in (1), the NMI is always between 0 and 1. The NMI is 1 if and only if the clusterings $U$ and $U'$ are identical. If the two clustering $U$ and $U'$ are mutually random, i.e. $P_{UU'}(c, c') = P_U(c)P_{U'}(c')$ ($P_U(c)$ and $P_{U'}(c')$ are independent events), then the NMI between them is 0. NMI thus negates the effect of randomness and makes for a reliable benchmark for cluster similarity across different cluster size distributions.

Supervised learning methods are evaluated by their accuracy. That is, given a labeled dataset (in which observations have assigned clusters), researchers evaluate how well the method can assign new observations to the given clusters. This differs fundamentally from the problem of unsupervised clustering, where the method is not trained on the ground-truth labels, but draws conclusions based on the mathematical properties of the data (e.g., its topology). In some cases, ground-truth labels may not even exist. As such, unsupervised clustering cannot be evaluated using traditional conceptions of accuracy.

Yet we still require the ability to compare clusterers, at least in cases where a ground-truth label is available. We utilize the NMI score for this purpose. It is like accuracy, in that it can provide insight into the amount of useful information captured by an unsupervised learning algorithm. Taking the NMI score between the ground-truth labels and the resultant clustering of an unsupervised technique, we can capture how accurately a clusterer matched the ground-truth without penalizing the technique for finding a different number of clusters. This application of the NMI score can be found in Section 5.1.

We further apply NMI scoring in order to assess cluster validity, which is distinct from adherence to ground-truth labels. We use the NMI to measure the stability of MapperPlus clusters relative to the choice of resolution and gain parameters. This is discussed in Section 4.3.

## 2.7 Datasets and availability

Four publicly available datasets with labeled clusters were used to compare MapperPlus to other commonly-used agnostic clustering algorithms. None of these datasets had missing values. The datasets are summarized in Table 2.

**Table 2. Summary of the publicly-available UCI Machine Learning Repository datasets used for method comparison.**

|  | Number of observations | Number of attributes | Known number of clusters |
|---|---|---|---|
| Wine | 178 | 13 | 3 |
| Breast Cancer | 569 | 32 | 2 |
| Iris | 150 | 4 | 3 |
| Rice | 3810 | 8 | 2 |

In a clinical case study examining the performance of MapperPlus for patient stratification, we applied the pipeline to a pediatric stem cell transplant dataset. This dataset comprised the treatment and outcomes for 187 pediatric patients with hematologic diseases who were subject to unmanipulated allogenic unrelated donor hematopoietic stem cell transplantation from 2000 to 2008 [23], [24]. Unlike the datasets used for comparison, this dataset is unlabelled. That is, there are no previously-defined classes that patients were assigned to. Further, this dataset contains a mix of both categorical and continuous variables, as well as pre-transplant data and post-transplant outcomes, which is characteristic of many medical datasets.

The patient cohort consists of 112 males and 75 females. In addition to patient sex, there are several categorical variables to describe the patients, such as disease type (acute myeloid leukemia, acute lymphocytic leukemia, chronic, nonmalignant, and lymphoma), disease category (malignant or nonmalignant), transplant compatibility statuses, and risk group (high or low). These categories, while informative to the conditions of the patients, were not considered labels as no one variable completely characterizes the patients. This can be contrasted to the Breast Cancer dataset, in which malignancy was the key characterization of each cell (observation).

We divided the 39 features available in the dataset into pre-transplant features and post-transplant outcomes. We identified 27 pre-transplant features that were input to MapperPlus. Categorical variables were translated into continuous values using frequency encoding [25]. The primary outcomes we evaluated were survival time (in days) and survival status following the procedure. Details regarding data processing and the handling of missing values can be found in the Supplement (S1 Text).

## 2.8 Statistical analysis

To measure feature and outcome differences among the clusters from the pediatric transplant dataset, we used chi-squared test (categorical) or ANOVA test (continuous). A $p$-value $< 0.05$ was considered statistically significant.

## 3 Results

### 3.1 Numerical validation: Comparison to other clustering methods

The clustering algorithms chosen for comparison were Affinity Propagation [26], DBSCAN [27], and MeanShift [28]. These were chosen due to their widespread use and because, like MapperPlus, they do not require an a priori knowledge of the number of clusters.

Each algorithm yields a list of cluster assignments on the dataset. The cluster assignments were then compared to the ground-truth labels of the dataset using NMI scores. For DBSCAN, we performed a manual tuning of the parameters. To make the fairest possible comparison, multiple parameter sets were considered and we reported the highest NMI score achieved.

Details on the Mapper inputs for each dataset can be found in the Supplement (S1 Text).

**Table 3. Performance of clustering algorithms on real-world data.** Shown are the NMI score and number of clusters (m′) predicted by MapperPlus, affinity propagation, DBSCAN, and Mean Shift on 4 real-world datasets. The datasets vary in the number of observations (N) and known numbers of clusters (m).

| Dataset | N | m | MapperPlus | | Affinity Propagation | | DBSCAN | | Mean Shift | |
|---|---|---|---|---|---|---|---|---|---|---|
| | | | NMI | m′ | NMI | m′ | NMI | m′ | NMI | m′ |
| Wine | 178 | 3 | 0.77 | 4 | 0.53 | 14 | 0.41 | 9 | 0.04 | 2 |
| Breast Cancer | 569 | 2 | 0.53 | 3 | 0.26 | 50 | 0.22 | 20 | 0.17 | 14 |
| Iris | 150 | 3 | 0.73 | 2 | 0.56 | 10 | 0.72 | 2 | 0.73 | 2 |
| Rice | 3810 | 2 | 0.57 | 2 | 0.19 | 97 | 0.03 | 28 | 0 | 0 |

We find that MapperPlus outperforms in NMI score to the other clustering algorithms (Table 3). In one dataset (Iris), DBSCAN produces a comparable result, with both algorithms detecting 2 clusters within the dataset. However, in the case of the breast cancer dataset, MapperPlus detected 3 clusters, which is an order of magnitude smaller than the number predicted by any of the other approaches. The result was similar for the rice dataset, where MapperPlus detected 2 clusters compared to affinity propagation detecting nearly 100. MapperPlus clustering is closer to the ground truth labels than the clustering returned by the other algorithms.

## 3.2 Using MapperPlus to predict survival in pediatric transplant patients

In a case study of MapperPlus in patient stratification, we analyzed the pediatric transplant dataset introduced in Section 2.7. The 27 pre-tranplant features (excluding outcome variables) were given to MapperPlus. Details regarding MapperPlus inputs for the clustering analysis can be found in the Supplement (S1 Text).

**3.2.1 Clustering analysis.** The MapperPlus algorithm yielded three disjoint clusters with $n_1 = 83$, $n_2 = 82$, and $n_3 = 22$.

As the goal of patient stratification is to yield clinically meaningful subgroups, we examined the characteristics of the clusters detected using statistical analysis. A summary of the analysis can be found in Tables 4–8. Of the 27 variables, 6 were categorical, but no data dictionary was included in the dataset. While these variables were input to MapperPlus, they were excluded from our statistical summaries.

MapperPlus partitioned the patient cohort into three groups in which 7 features had mean values that were statistically significantly different. (Two of these features had incomplete labeling and are thus not summarized in the associated tables). In addition to this phenotypic clustering, MapperPlus was able to separate the patients by survival rate. Cluster 1 patients had an above-average survival rate (61.45% survival, $p = 0.03$), whereas the overall group survival rate is 54.6%. Cluster 2 patients had the worst survival rate at 43.9% ($p = 0.03$) and Cluster 3 has the best survival rate at 68.2% ($p = 0.03$). In addition, Cluster 2 patients had the shortest average survival time (in days) in the first year of 259.02 while Cluster 1 had the longest average survival time of 312 days in the first year ($p = 0.035$).

## 3.3 Cluster validity: Choice of resolution and gain

It is well known that Mapper output can change significantly relative to small changes in its input parameters [29]. While some results guiding stability for resolution and gain choices have been found in simplified instances [14], there are currently no general guidelines for optimal parameter choice. Since MapperPlus leverages the Mapper graph directly, it too is affected by the instability of the Mapper algorithm.

**Table 4. Summary of binary and continuous pre-transplant variables, outcome variables.** For continuous variables, we include the mean and standard deviation for the overall dataset and clusters. For binary variables, we include the count.

| | All (n = 187) | N1 (n = 83) | N2 (n = 82) | N3 (n = 22) | p-value |
|---|---|---|---|---|---|
| **Pre-transplant Features** | | | | | |
| Donor age | 33.47 ± 8.27 | 32.82 ± 9.27 | 31.61 ± 8.55 | 31.70 ± 6.83 | 0.22 |
| Recipient age | 9.93 ± 5.31 | 5.61 ± 2.71 | 14.19 ± 3.79 | 10.35 ± 4.54 | **<0.001** |
| CD34+ cell doses per kilogram (10^6/kg) | 11.89 ± 9.91 | 15.03 ± 10.44 | 8.53 ± 7.53 | 12.56 ± 11.94 | **<0.001** |
| CD3+ cell doses per kilogram (10^6/kg) | 4.74 ± 3.86 | 6.11 ± 4.50 | 3.15 ± 2.75 | 4.43 ± 2.87 | **<0.001** |
| CD3+ cell to CD34+ cell ratio | 5.39 ± 9.60 | 5.16 ± 11.17 | 5.61 ± 8.80 | 4.16 ± 3.69 | 0.81 |
| Recipient stem cell body mass (kg) | 35.81 ± 19.65 | 20.83 ± 9.14 | 50.27 ± 18.01 | 35.08 ± 15.02 | **<0.001** |
| Donor age below 35 | 104 (55.61%) | 48 (57.83%) | 40 (48.78%) | 16 (72.73%) | 0.115 |
| Presence of cytomegalovirus infection (CMV) in donor prior to transplantation | 72 (38.50%) | 33 (39.76%) | 30 (36.59%) | 9 (69.23%) | 0.889 |
| Recipient age below 10 | 99 (52.94%) | 83 (100%) | 7 (8.54%) | 9 (40.91%) | **0.001** |
| Recipient gender | 112 males (65.24%) | 46 (55.42%) | 52 (63.41%) | 14 (63.64%) | 0.537 |
| Presence of CMV in recipient prior to transplantation | 100 (53.48%) | 42 (50.60%) | 44 (53.66%) | 14 (63.64%) | 0.552 |
| Disease categorized as malignant | 155 (82.89%) | 62 (74.70%) | 75 (91.46%) | 18 (81.82%) | **0.017** |
| Compatibility of donor and recipient according to gender (incidence of female to male) | 32 (17.11%) | 11 (13.25%) | 16 (19.51%) | 5 (22.73%) | 0.429 |
| Compatibility of donor and recipient according to blood group (incidence of matched) | 52 (27.81%) | 21 (25.30%) | 26 (31.71%) | 6 (27.27%) | 0.655 |
| Incidence of HLA match | 159 (85.03%) | 82 (98.80%) | 62 (75.61%) | 15 (68.18%) | **<0.001** |
| Risk Group (high, low) | 69 high risk (36.90%) | 26 (31.33%) | 34 (41.47%) | 9 (40.91%) | 0.369 |
| Stem cells source (peripheral, bone marrow) | 145 peripheral (77.54%) | 67 (80.72%) | 60 (73.17%) | 18 (81.82%) | 0.446 |
| **Outcomes** | | | | | |
| 1-year Survival Time (Days) | 286 ± 132.39 | 312.11 ± 116.25 | 259.02 ± 143.87 | 291.72 ± 130.60 | **0.035** |
| 1-year Survival Rate | 64.71% | 74.39% | 54.88% | 68.18% | **0.041** |
| Study Duration Survival Rate | 54.54% | 61.45% | 43.90% | 68.18% | **0.03** |

Given that resolution and gain are tunable, it is necessary that we have a rigorous and reproducible method for selecting these parameters. In the absence of ground-truth labeling (as expected from many patient datasets and represented here in our pediatric transplant case study), we use stability as a guide for parameter selection. We select resolution $r$ and gain $g$ pairs such that the resultant clustering does not change significantly due to small changes in $r$ and $g$, as measured by the NMI score.

We apply NMI to the problem of finding stable parameter choices as follows. For a given resolution $r$ ang $g$, we represent the associated MapperPlus clustering $U^{(r, g)}$ as a point. The pairwise NMI scores are then averaged over the clusterings of all neighbors (the 8 adjacent

**Table 5. This table presents a summary of the variable of age in years discretized into intervals.** The overall row shows the distribution (counts) in each age interval in the entire dataset. The remaining rows show the distribution for each cluster. This variable was significant with $p < 0.001$.

| | (0,5] | (5,10] | (10,20] |
|---|---|---|---|
| Overall | 47 | 51 | 89 |
| Cluster 1 | 38 | 45 | 0 |
| Cluster 2 | 6 | 1 | 75 |
| Cluster 3 | 3 | 5 | 14 |

**Table 6. This table presents a summary of the variable of disease category (acute lymphocytic leukemia (ALL), acute myelogenous leukemia (AML), chronic disease, non-malignant disease, or lymphoma).** The overall row shows the distribution (counts) of disease in the entire dataset. The remaining rows show the distribution for each cluster. The variable was significant with $p < 0.010$.

|  | ALL | AML | Chronic | Non-malignant | Lymphoma |
|---|---|---|---|---|---|
| Overall | 68 | 33 | 45 | 32 | 9 |
| Cluster 1 | 34 | 7 | 19 | 21 | 2 |
| Cluster 2 | 26 | 20 | 22 | 7 | 7 |
| Cluster 3 | 8 | 6 | 4 | 4 | 0 |

points) of the point $(r, g)$ to yield a mean cluster score. We set a mean cluster threshold and only consider $(r, g)$ pairs with average scores greater than that threshold. This ensures that the clusters we examine are relatively stable, i.e., small perturbations in the parameter choice of Mapper result in small or no change in the final clustering as measured by NMI. This method is only possible because MapperPlus returns disjoint clusters from the Mapper graph. See the Supplement (S1 Text) for further details regarding NMI scoring in this study.

Stability can inform our choice of tunable parameters to an extent, but we must also consider the degree of resolution we want from the dataset. Very high resolution will separate points to a great degree, naturally leading to a larger number of clusters each containing few observations. Conversely, very high gain brings points close together, leading to a smaller number of clusters. MapperPlus offers much-needed flexibility with the tuning of the resolution and gain parameters.

## 4 Discussion

The main contribution of our work is the development of a ready-to-use topological pipeline for clustering high-dimensional data. Our pipeline is an agnostic approach that uses topological data analysis to identify stable disjoint clusters by overcoming the limitations of early developments in this field. We have shown the performance of this new pipeline in multiple datasets with benchmarks (higher NMI scoring and number of detected clusters being closer to ground-truth labels) that suggest widespread applicability. Lastly, we present a case-in-point of how our pipeline may be used in the field of precision medicine using publicly-available clinical data that has no ground-truth labeling towards the expected number of clinically meaningful subgroups.

Our clustering pipeline starts with Mapper, a TDA technique that leverages the topological information inherent to the structure of data to produce a graph. Previous studies have shown how Mapper can identify subsets of data that may not be easily found using traditional clustering methods [29]. Unlike traditional methods, Mapper preserves the local connectivity between clusters, which allows for the exploration of complex relationships in datasets. Its

**Table 7. This table presents a summary of the serological compatibility of the donor and recipient according to CMV prior to transplantation.** The higher the value (0—3), the lower the compatibility. The overall row shows the distribution (counts) for patients in the entire dataset. The remaining rows show the distribution for each cluster. The p-value was $p = 0.831$.

|  | 0 | 1 | 2 | 3 |
|---|---|---|---|---|
| Overall | 48 | 27 | 57 | 39 |
| Cluster 1 | 32 | 12 | 22 | 17 |
| Cluster 2 | 26 | 13 | 27 | 16 |
| Cluster 3 | 6 | 2 | 8 | 6 |

**Table 8. This table presents a summary of the HLA matches across cluster (10/10, 9/10 are considered matches).** The overall row shows the distribution (counts) for the patients in the entire dataset. The remaining rows show the distribution for each cluster. The p-value was $p < 0.001$.

|  | 10/10 | 9/10 | 8/10 | 7/10 |
|---|---|---|---|---|
| Overall | 94 | 65 | 23 | 5 |
| Cluster 1 | 44 | 38 | 1 | 0 |
| Cluster 2 | 36 | 26 | 15 | 5 |
| Cluster 3 | 14 | 1 | 2 | 0 |

approach is also hypothesis-free: it requires no insight into the number of clusters in the dataset to create the graph. Yet Mapper falters as it does not yield disjoint clusters, but a graph. Current approaches for resolving this issue often rely on the exclusion of patients and their valuable data, as seen for example in [19], which may introduce bias.

MapperPlus directly addresses this issue by extending the Mapper algorithm to yield disjoint clusters. As depicted in Fig 1, a Mapper graph is generated in Steps 1—4. In Step 5, WLCF is applied to the Mapper graph to automatically detect the number of communities in the graph. This yields an overlapping clustering on the dataset (Step 6), which is a direct result of the structure of the Mapper graph. But MapperPlus also goes beyond the partitioning of the Mapper graph in order to rigorously detect disjoint clusters. It does so by generating a novel network of instances where each node is an individual observation (Step 7). Weighted edges connect nodes that are binned together in the original Mapper graph. This crucial step embeds the topological information of the Mapper graph while neatly avoiding the overlapping binning problem. Finally, in Step 8, WLA is applied to the novel graph using the number of communities detected originally. This yields a rigorous disjoint clustering of the Mapper graph that preserves all the topological information. Thus, MapperPlus presents a new solution to the pressing question of identifying disjoint clusters using Mapper.

MapperPlus also provides particular advantages as a clustering technique for precision medicine. Specifically, it directly addresses the three key challenges of patient stratification: 1) the unknown number of clusters, 2) the need for assessing cluster validity, and 3) the clinical interpretability. In applications to publicly available data, we demonstrate that the MapperPlus pipeline directly addresses all three of these issues.

First, MapperPlus is a wholly agnostic clustering technique. It does not require any knowledge regarding the number of clusters within a dataset in order to function. This type of agnostic approach is uniquely suited to the needs of patient stratification, particularly when studying complex and heterogeneous diseases where the number of patient subtypes may be unknown.

In addition, MapperPlus outperforms three other agnostic clustering algorithms (affinity propagation, DBSCAN, and mean shift) when tested on four labeled datasets. The datasets varied in size (from 178 observations to 3810) and in number of attributes (from 4 to 32). It was important to use datasets with known labels to evaluate the performance of MapperPlus. MapperPlus consistently achieved higher NMI scores (a metric of adherence to ground-truth) than these leading clustering algorithms, often by a significant margin (Table 3). MapperPlus clustering consistently predicted fewer clusters and the resultant clusters had greater similarity to ground truth labels than the other algorithms. In the rice dataset, MapperPlus returned 2 clusters, while affinity propagation and DBSCAN predicted 97 and 28 clusters, respectively. In the breast cancer dataset, MapperPlus returned 3 clusters, while affinity propagation and DBSCAN predicted 50 and 20, respectively. This large number of clusters can lead to problems of interpretability. First, it can be challenging to assign meaning to so many clusters. Second, if

the number of observations in a cluster is small, statistical comparison to other clusters may not be possible. MapperPlus can be tuned to detect meaningful differences between observations while avoiding returning such a large number of clusters, which is integral to ensuring that the generated clusters can be assigned clinical meaning.

Second, we provide a method for new assessing cluster stability and thereby selecting the tunable MapperPlus parameters. We need to be confident that our clustering is stable if we wish to base clinical decisions on it. There is currently no consensus on how to best select resolution and gain values. A major contribution of our work is a heuristic for selecting parameters: the NMI score. Selecting gain and resolution parameters that produce NMI scores that exceed a set benchmark ensures that the clusters are stable with respect to the choice of parameters. The choice of an appropriate benchmark is dependent on the features of the dataset being studied. In the absence of theoretical assurances of stability, the NMI score offers a method for evaluating the stability of clusters.

After evaluating MapperPlus with datasets with known labels, we turned to analyzing a publicly available cell transplantation data. This dataset is similar to many medical datasets, having a mix of continuous and categorical features and, importantly, no labeled clusters. The goal was to determine if the patients within the clusters returned by MapperPlus had discernibly similar clinical features. MapperPlus detected three clusters within the data. 7 of the pre-transplant variables were statistically significant across the clusters. These clusters also had significantly different survival rates and times, which is suggestive of patient stratification for risk. Thus, MapperPlus clustering appears to capture clinical insights from the data through the detection of meaningful features that could differentiate outcomes across clusters. While further studies are needed, these insights may be highly informative in the future determination of what features relate to improved survival in patients.

The use of publicly available datasets to evaluate the performance of MapperPlus and other clustering algorithms ensures transparency and the possibility of evaluation by others. Our study of the transplant data was limited by the lack of a completed data dictionary, so our statistical summary was incomplete. The transplant data did not have a complete data dictionary, so our statistical summary was incomplete. The second limitation is the open question of tuning parameters for MapperPlus. While we can provide some guidelines for parameter choice, theoretical guarantees of stability for MapperPlus remains an open area of research requiring more development. Finally, the MapperPlus pipeline is a collective of new and previously-established analytical components. Some of the intermediate components (e.g. the clusterer) are used independently to observations, which can require user-selected parameters. As such, our pipeline may not be considered wholly agnostic. This is most notable in Step 3 of our pipeline, where user-selected parameters are used for partial clustering on the dataset relative to the cover but not for the final clustering of observations. In future work, we would like to explore the use of a more agnostic intermediate clusterer.

In conclusion, MapperPlus is an applied topological tool capable of clustering data in an agnostic manner with valid and meaningful interpretation, which facilitates precision medicine applications.

## Supporting information

**S1 Text. Supplementary material.**
(PDF)

## Author Contributions

**Conceptualization:** Esha Datta, Aditya Ballal, Javier E. López, Leighton T. Izu.

**Data curation:** Esha Datta.

**Formal analysis:** Esha Datta, Aditya Ballal, Leighton T. Izu.

**Funding acquisition:** Leighton T. Izu.

**Investigation:** Esha Datta, Aditya Ballal, Leighton T. Izu.

**Methodology:** Esha Datta, Aditya Ballal, Leighton T. Izu.

**Project administration:** Leighton T. Izu.

**Resources:** Leighton T. Izu.

**Software:** Esha Datta, Aditya Ballal.

**Supervision:** Javier E. López, Leighton T. Izu.

**Validation:** Esha Datta, Aditya Ballal, Leighton T. Izu.

**Writing – original draft:** Esha Datta, Aditya Ballal, Leighton T. Izu.

**Writing – review & editing:** Esha Datta, Aditya Ballal, Javier E. López, Leighton T. Izu.

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
