## [Decision Letter · Decision Letter 0]

11 Jan 2023

PDIG-D-22-00336

MapperPlus: Agnostic Clustering of High-Dimension Data for Precision Medicine

PLOS Digital Health

Dear Dr. Izu,

Thank you for submitting your manuscript to PLOS Digital Health. After careful consideration, we feel that it has merit but does not fully meet PLOS Digital Health's publication criteria as it currently stands. Therefore, we invite you to submit a revised version of the manuscript that addresses the points raised during the review process.

The reviewers raised some concerns regarding the manuscript in its current version. Please address their comments.

Please submit your revised manuscript within 60 days Mar 12 2023 11:59PM. If you will need more time than this to complete your revisions, please reply to this message or contact the journal office at digitalhealth@plos.org. Please include the following items when submitting your revised manuscript:

We look forward to receiving your revised manuscript.

Kind regards,

Nadav Rappoport, Ph.D.

Academic Editor

PLOS Digital Health

Journal Requirements:

1. Please send a completed 'Competing Interests' statement, including any COIs declared by your co-authors. If you have no competing interests to declare, please state "The authors have declared that no competing interests exist". Otherwise please declare all competing interests beginning with the statement "I have read the journal's policy and the authors of this manuscript have the following competing interests:"

a. State what role the funders took in the study. If the funders had no role in your study, please state: “The funders had no role in study design, data collection and analysis, decision to publish, or preparation of the manuscript.”

b. If any authors received a salary from any of your funders, please state which authors and which funders.

3. We ask that a manuscript source file is provided at Revision. Please upload your manuscript file as a .doc, .docx, .rtf or .tex.

4. Please provide separate figure files in .tif or .eps format only and remove any figures embedded in your manuscript file. Please also ensure that all files are under our size limit of 10MB.

Additional Editor Comments (if provided):

Reviewers' comments:

Reviewer's Responses to Questions

**Comments to the Author**

1. Does this manuscript meet PLOS Digital Health’s publication criteria? Is the manuscript technically sound, and do the data support the conclusions? The manuscript must describe methodologically and ethically rigorous research with conclusions that are appropriately drawn based on the data presented.

Reviewer #1: Partly

Reviewer #2: No

Reviewer #3: Yes

2. Has the statistical analysis been performed appropriately and rigorously?

Reviewer #1: N/A

Reviewer #2: No

Reviewer #3: Yes

3. Have the authors made all data underlying the findings in their manuscript fully available (please refer to the Data Availability Statement at the start of the manuscript PDF file)?

Reviewer #1: Yes

Reviewer #2: Yes

Reviewer #3: Yes

4. Is the manuscript presented in an intelligible fashion and written in standard English?

Reviewer #1: Yes

Reviewer #2: Yes

Reviewer #3: Yes

5. Review Comments to the Author

Reviewer #1: Review

MapperPlus Agnostic Clustering of High-Dimension Data for Precision Medicine

The article deals with a new unsupervised approach to support personalized medicine decisions (precision medicine). The authors suggest using clustering algorithm that assumes low error clustering (agnostic clustering). 

Comments:

1) The term sub-clusters used in the text is confusing. We are talking about clusters, that are homogeneous population sub-groups. We are not dividing clusters to sub-clusters. 

2) The literature review starts with the precision medicine requirements and challenges and continues with cluster analysis challenges. Despite that the text in the Introduction part is well constructed and clear, most of the facts and statements are not based on any scientific literature. At least, they are not cited. For example, in the description of 2-nd, 3-rd and 4-th challenges of the use of cluster analysis methods there is not a single citation. The literature review must be significantly expanded. The authors must refer to additional aspects, such as interpretation and multidimensionality in a medical context. Two parts of the Introduction part must be connected in the context of the research topic. 

3) In the Materials and Methods section authors must provide the Notations table used in the paper and all databases' characteristics. 

4) MapperPlus approach must be presented in the Methods section and not in the Results section. 

5) The authors present the pediatric patients' dataset as a main dataset, but later use 4 different datasets, part of them is not appropriate for the "precision medicine" goal. For example, Iris dataset is not a good example of multivariate data with hidden subpopulations. The Wine dataset has a very small number of independent variables and standard classification techniques provide very high accuracy. The use of cluster analysis approaches is justified when we can assume that there are hidden sub-groups in the population that can be identified by the unsupervised techniques. I suggest using at lest 3 multivariate medical datasets with a poor initial classification accuracy. 

6) MapperPlus results must be compared to other clustering techniques with the same number of clusters. If the novel method will provide better results, at least in some of the datasets, it can be defined as a successful and usable approach. 

7) The main goal of the research is announced as a novel approach to support the precision medicine. But the main part of the discussion addresses the methodological improvement of the existing clustering methodology. The connection with a medical performance is not emphasized enough.

Reviewer #2: This paper presents an agnostic clustering of high-dimension data for precision medicine. The novelty of this work seems marginal. A clustering method with eight steps is devised. But the significance of the combination of these steps (when compared with the state-of-the-art clustering methods) remains to be justified. Moreover, the precision medicine is a very wide area. It is also unclear for what types of precision medicine data the proposed method is designed.

Reviewer #3: This article analyses the development of an unsupervised cluster pipeline which deals with the issue of deciding the number of clusters, their stability and quality. This article is well written, with a sound methodology, and the conclusions are derived from the results. The results of this study seem helpful to the scientific community. However, some issues should be solved and some information clarified.

The main issue is that the description of the study populations is almost absent. There is no description of missing data. Were there missing data, and how did the algorithm deal with that? Or did the researchers exclude missing information (complete analysis), or they applied some imputation technique before the clustering? 

Other minor issues are the following:

It is unclear whether the algorithm normalised the data or whether it should have been normalised before the application.

The Cluster quality assessment using NMI scores is only possible if the data are labelled. If the pipeline has an agnostic approach (not labelled), how can NMI be applied for selecting parameters?

The article describes clusters and subclusters. Is there any difference?

Can the clustering method be parametrised?

Define what TDA is

Is the lens selection dependent on the hyper-parameters of the algorithm?

Can the clustering method be parametrised?

6. PLOS authors have the option to publish the peer review history of their article (what does this mean?). If published, this will include your full peer review and any attached files.

**Do you want your identity to be public for this peer review?** For information about this choice, including consent withdrawal, please see our Privacy Policy.

Reviewer #1: No

Reviewer #2: No

Reviewer #3: No

---

## [Decision Letter · Decision Letter 1]

2 May 2023

PDIG-D-22-00336R1

MapperPlus: Agnostic Clustering of High-Dimension Data for Precision Medicine

PLOS Digital Health

Dear Dr. Izu,

Thank you for submitting your manuscript to PLOS Digital Health. After careful consideration, we feel that it has merit but does not fully meet PLOS Digital Health's publication criteria as it currently stands. Therefore, we invite you to submit a revised version of the manuscript that addresses the points raised during the review process.

Please submit your revised manuscript within 30 days Jun 01 2023 11:59PM. If you will need more time than this to complete your revisions, please reply to this message or contact the journal office at digitalhealth@plos.org. Please include the following items when submitting your revised manuscript:

We look forward to receiving your revised manuscript.

Kind regards,

Nadav Rappoport, Ph.D.

Academic Editor

PLOS Digital Health

Journal Requirements:

Additional Editor Comments (if provided):

Reviewers' comments:

Reviewer's Responses to Questions

**Comments to the Author**

1. If the authors have adequately addressed your comments raised in a previous round of review and you feel that this manuscript is now acceptable for publication, you may indicate that here to bypass the “Comments to the Author” section, enter your conflict of interest statement in the “Confidential to Editor” section, and submit your "Accept" recommendation.

Reviewer #1: All comments have been addressed

Reviewer #2: All comments have been addressed

2. Does this manuscript meet PLOS Digital Health’s publication criteria? Is the manuscript technically sound, and do the data support the conclusions? The manuscript must describe methodologically and ethically rigorous research with conclusions that are appropriately drawn based on the data presented.

Reviewer #1: Yes

Reviewer #2: Yes

3. Has the statistical analysis been performed appropriately and rigorously?

Reviewer #1: Yes

Reviewer #2: Yes

4. Have the authors made all data underlying the findings in their manuscript fully available (please refer to the Data Availability Statement at the start of the manuscript PDF file)?

Reviewer #1: Yes

Reviewer #2: Yes

5. Is the manuscript presented in an intelligible fashion and written in standard English?

Reviewer #1: Yes

Reviewer #2: Yes

6. Review Comments to the Author

Reviewer #1: After the revision, the article is much more clear and more organized. Most of the comments were fully addressed. 

I have three additional comments:

In line 5 in the abstract must be an "unknown number of clusters". Clusters in the plural. 

In line 10 in the abstract must be "key accuracy/performance metrics". Just "key metrics" is not clear. 

My main concern (again) addresses the databases chosen. I understand the authors' explanation that 3 of 4 databases were chosen in order to justify the use of the novel algorithm. The hidden classes are known in advance and it is convenient to use these databases for this reason. I still think that if the main motivation of your method is to improve the functionality of precision medicine, you have to demonstrate it in the relevant field. You can find databases that include hidden classes in advance. At least try to demonstrate the MapperPlus on 2-3 databases with different levels of data complication.

Reviewer #2: The authors have addressed my previous concerns. I think it will be ready for acceptance provided that some minor issues are further tackled.

1. The proposed clustering algorithm is quite complicated, which consists of eight steps. A clear summarization about how each of these eight steps contributes to the final performance is encouraged.

2. The ensemble clustering is able to combine multiple clustering results into a more robust clustering. Some ensemble clustering works, such as "Locally weighted ensemble clustering", "Enhanced Ensemble Clustering via Fast Propagation of Cluster-wise Similarities", and "Fast Multi-view Clustering via Ensembles: Towards Scalability, Superiority, and Simplicity" can also be considered in the future extensions.

7. PLOS authors have the option to publish the peer review history of their article (what does this mean?). If published, this will include your full peer review and any attached files.

**Do you want your identity to be public for this peer review?** For information about this choice, including consent withdrawal, please see our Privacy Policy. 

Reviewer #1: No

Reviewer #2: No

---

## [Decision Letter · Decision Letter 2]

25 Jun 2023

MapperPlus: Agnostic Clustering of High-Dimension Data for Precision Medicine

PDIG-D-22-00336R2

Dear Dr. Izu,

We are pleased to inform you that your manuscript 'MapperPlus: Agnostic Clustering of High-Dimension Data for Precision Medicine' has been provisionally accepted for publication in PLOS Digital Health.

Best regards,

Nadav Rappoport, Ph.D.

Academic Editor

PLOS Digital Health

Reviewer Comments (if any, and for reference):

Reviewer's Responses to Questions

**Comments to the Author**

1. If the authors have adequately addressed your comments raised in a previous round of review and you feel that this manuscript is now acceptable for publication, you may indicate that here to bypass the “Comments to the Author” section, enter your conflict of interest statement in the “Confidential to Editor” section, and submit your "Accept" recommendation.

Reviewer #1: All comments have been addressed

Reviewer #2: All comments have been addressed

2. Does this manuscript meet PLOS Digital Health’s publication criteria? Is the manuscript technically sound, and do the data support the conclusions? The manuscript must describe methodologically and ethically rigorous research with conclusions that are appropriately drawn based on the data presented.

Reviewer #1: Yes

Reviewer #2: Yes

3. Has the statistical analysis been performed appropriately and rigorously?

Reviewer #1: Yes

Reviewer #2: Yes

4. Have the authors made all data underlying the findings in their manuscript fully available (please refer to the Data Availability Statement at the start of the manuscript PDF file)?

Reviewer #1: Yes

Reviewer #2: Yes

5. Is the manuscript presented in an intelligible fashion and written in standard English?

Reviewer #1: Yes

Reviewer #2: Yes

6. Review Comments to the Author

Reviewer #1: The recent update looks much better and the article is ready to be published.

Good luck!

Reviewer #2: I have no further comments.

7. PLOS authors have the option to publish the peer review history of their article (what does this mean?). If published, this will include your full peer review and any attached files.

**Do you want your identity to be public for this peer review?** For information about this choice, including consent withdrawal, please see our Privacy Policy.

Reviewer #1: No

Reviewer #2: None
